# Microstructure Characteristics and Hydrogen Storage Kinetics of Mg_77+*x*_Ni_20−*x*_La_3_ (*x* = 0, 5, 10, 15) Alloys

**DOI:** 10.3390/ma16134576

**Published:** 2023-06-25

**Authors:** Hongxiao Tian, Qichang Wang, Xia Li, Long Luo, Yongzhi Li

**Affiliations:** 1School of Science, Inner Mongolia University of Science and Technology, Baotou 014010, China; tianhongxiao@imust.edu.cn (H.T.);; 2Baotou Materials Research Institute of Shanghai Jiao Tong University, Baotou 014010, China

**Keywords:** Mg-based alloy, lanthanum, microstructure, hydrogen storage kinetic

## Abstract

Mg_77+*x*_Ni_20−*x*_La_3_ (*x* = 0, 5, 10, 15) alloys were successfully prepared by the vacuum induction melting method. The structural characterizations of the alloys were performed by using X-ray diffraction and scanning electron microscope. The effects of nickel content on the microstructure and hydrogen storage kinetic of the as-cast alloys were investigated. The results showed that the alloys are composed of a primary phase of Mg_2_Ni, lamella eutectic composites of Mg + Mg_2_Ni, and some amount of LaMg_12_ and La_2_Mg_17_. Nickel addition significantly improved the hydrogen-absorption kinetic performance of the alloy. At 683 K, Mg_77_Ni_20_La_3_ alloy and Mg_82_Ni_15_La_3_ alloy underwent hydrogen absorption and desorption reactions for 2 h, respectively, and their hydrogen absorption and desorption capacities were 4.16 wt.% and 4.1 wt.%, and 4.92 wt.% and 4.69 wt.%, respectively. Using the Kissinger equation, it was calculated that the activation energy values of Mg_77_Ni_20_La_3_, Mg_82_Ni_15_La_3_, Mg_87_Ni_10_La_3_ and Mg_92_Ni_5_La_3_ alloys were in the range of 68.5~75.2 kJ/mol, much lower than 150~160 kJ/mol of MgH_2_.

## 1. Introduction

Fossil fuels, the main energy source for the past two centuries, have caused energy crisis and environmental pollution problems due to excessive consumption [1,2,3]. Therefore, finding new energy sources is urgent [4]. Hydrogen is a clean energy source with many advantages such as versatility, high utilization efficiency, environmental compatibility, and inexhaustible [5]. Hydrogen energy economy involves hydrogen production, storage, transportation and application. Among them, hydrogen storage and transportation are the core and bottleneck technologies [6]. To achieve large-scale applications, the hydrogen storage system needs to be safe, efficient, and cost-effective [7]. Traditional gas storage, which can be pressurized up to 70 MPa with technical progress, pose considerable safety risk for mobile use [8]. Cryogenic liquid hydrogen storage, which consume more than 30% calorific value of the liquid hydrogen, are not suitable for popularization in the civil field [9]. Solid-state hydrogen storage materials are the first choice for large-scale hydrogen storage systems because of their safety, high capacity, and low pressure [10,11,12,13]. Magnesium-based alloys are the most viable solid-state hydrogen storage materials, owning to their low cost, high hydrogen storage capacity, high cycling stability and high safety [14,15,16,17]. However, the slow kinetics and high thermodynamics hinder their development [18].

A great deal of experimental researchers have been conducted in the last few decades to address the drawback of Mg-based alloys, such as alloying [19,20,21], nano-crystallization [22,23,24], doping catalyst [25,26,27] and exploring new treatment methods [28,29,30], etc. By alloying various transition metals (TM) with Mg, TM was found to reduce the thermal stability of MgH_2_ and enhance the hydrogen reaction kinetics of MgH_2_ [31,32]. Mg-Si alloys formed by alloying Mg and Si elements [33] can significantly reduce the thermal stability of the alloy. When Mg is alloyed with Ni elements, the hydrogen absorption/desorption phases transform from Mg/MgH_2_ to Mg_2_Ni/Mg_2_NiH_4_, and the dehydrogenation enthalpy decreases from 76 kJ/mol H_2_ to 64 kJ/mol H_2_ [34,35,36]. Ding et al. reported that the Mg-Mg_2_Ni eutectic with a large amount of phase/grain boundaries is beneficial for the dissociation of H_2_ and permeation of hydrogen atoms [34]. In addition, the enthalpy of hydrogen desorption of the Mg-In alloy [37] is reduced to 67.8 kJ/mol, which is very close to the Mg_2_Ni system. The TM elements can be used as a special catalyst to lower the energy barrier for the recombination of hydrogen atoms and the decomposition of hydrogen molecules [38]. However, the Mg-TM alloys possess better hydrogen absorption/desorption kinetics with the expense of reversible hydrogen storage capacity [39,40].

Beside of TM, rare earth (RE) metals were also used widely for producing Mg-based alloys to enhance the hydrogenation kinetics. The in situ formed nano-scale rare earth hydrides (REH*_x_*) bring interfacial channels and therefore facilitates the diffusion of hydrogen in the alloy [41,42,43]. Moreover, REH*_x_* has a unique “hydrogen pumping” function, which can effectively reduce the apparent activation energy [44,45,46]. According to the Mg-La phase diagram [47], it is known that Mg-LaMg*_x_* eutectic can be formed even at very high Mg content. Lass et al. [48] found that after alloying with La, the reaction enthalpy of MgH_2_ and Mg_2_NiH_4_ in the Mg-Ni alloy were reduced by 8 kJ/mol and 5 kJ/mol, respectively. REH*_x_* in Mg-RE-TM alloys, which can be uniformly dispersed at the grain boundaries of the Mg matrix, effectively reduces the catalytic dead zone and improves the catalytic efficiency [49]. In addition, the stability of the nano phase structure of Mg-based alloys can be improved by pinning of REH*_x_* to the Mg nano grain boundaries, which inhibits the growth of Mg/MgH_2_ grains [50].

The balance between improving of reaction kinetics and decreasing of hydrogen storage capacity should be investigated carefully when increasing the proportion of Ni or RE in Mg-Ni-RE alloys. In the present study, Mg_77+*x*_Ni_20−*x*_La_3_ (*x* = 0, 5, 10, 15) alloys were prepared by induction melting method. The microstructure, phase transformation and hydrogen storage kinetic of the alloys with different Mg and Ni ratios are studied. The synergistic catalytic effect of LaH_3.05_ phase and Mg_2_Ni phase on the hydrogen absorption and desorption process of the alloy was analyzed.

## 2. Materials and Methods

### 2.1. Material Preparation Methods

Alloys with the nominal composition of Mg_77+*x*_Ni_20−*x*_La_3_ (*x* = 0, 5, 10, 15) were made from the appropriate amounts of pure magnesium, nickel, and lanthanum by using vacuum induction melting. The purity of the starting materials was not less than 99.5%, and the overall weight of the alloy was ~5 kg. Pure argon was used to fill in the furnace to prevent oxidation. The metallic solution was poured into an iron mold and cooled to indoor temperature. Afterward, the alloy ingots were mechanically pulverized into powder samples with a particle size of <200 mesh.

### 2.2. Structural Characterizations

The physical composition of the alloy was analysed by X-ray powder diffraction (XRD). The test instrument was a PANalytical X’pert Powder X-ray diffractometer from Panacor, Eindhoven, The Netherlands, using a Cu Kα target, a tube voltage of 40 kV, a current of 150 mA, sampling in a continuous scan mode, a scan speed of 2°/min, a step size of 0.02° and a scan range of 10–80°. To determine the phase composition of the alloy, XRD patterns were analysed using X’Pert HighScore Plus software, V3.8. After the ingot sections had been ground and polished in alcohol, the morphological structure of the ingot sections was observed using a TESCAN GAIA-3 (Tescan, Brno, Czech Republic) field emission scanning electron microscope (SEM) and the phase composition of the different morphological micro-zones was analysed using the energy spectrum analysis (EDS) supplied with the equipment. The block and powdered samples are glued to the metal base with conductive adhesive and the SEM is performed on the block before hydrogen absorption.

### 2.3. Hydrogen Storage Measurements

Hydrogen sorption kinetic tests were performed in a Sievert’s apparatus under high-purity hydrogen (99.999% purity). Each sample weighed ~200 mg. The reactor was heated by a resistance furnace with an accuracy of ±0.5 K. Before measuring the hydrogenation kinetics, the samples were activated to three hydrogen absorption and desorption cycles at 653 K. Isothermal hydrogen absorption kinetic measurements were made in the temperature range from 473 to 683 K and with 3 MPa of hydrogen pressure. Calorimetric measurements of the full hydrogenated samples were also made using a NETZSCH (Selb, Germany) DSC 204 HP differential scanning calorimeter (DSC). The samples were heated from indoor temperature to 750 K with a heating rate of 5, 10, 20 and 40 K/min. The tests were performed under a pure argon flow (50 mL/min).

## 3. Results

### 3.1. Microstructure Characteristics

The X-ray diffraction patterns (XRD) of Mg_77+*x*_Ni_20−*x*_La_3_ (*x* = 0, 5, 10, 15) alloys with four different Mg and Ni contents (*x* = 0, 5, 10, 15) are shown in Figure 1. The alloys before hydrogenation mainly consist of four phases: LaMg_12_, Mg_2_Ni, La_2_Mg_17_ and Mg. As the Mg content increases, the intensities of LaMg_12_, Mg_2_Ni and La_2_Mg_17_ phases gradually decrease for Mg_82_Ni_15_La_3_ alloy. It is speculated that this is because the Mg content in the Mg phase increases, while the Mg content in the LaMg_12_, Mg_2_Ni and La_2_Mg_17_ phases decreases accordingly. Therefore, the intensities of the diffraction peaks of LaMg_12_, Mg_2_Ni and La_2_Mg_17_ phases decrease gradually. The grain boundaries formed between the multi-phase structures can provide channels for the diffusion of hydrogen sites and hydrogen atoms in the alloy. The defects and dislocations on the grain boundaries can lower the diffusion energy barrier of hydrogen atoms and enhance the diffusion coefficient of hydrogen atoms, thus accelerating the migration of hydrogen atoms in the alloy, which is conducive to improving the kinetic properties of the hydrogen absorption and desorption processes in the alloy. The C peak in the alloy is caused by the graphite crucible used to prepare the alloy in our experiment. The graphite crucible reacts with some metals at high temperatures to produce carbides, which are transferred to the alloy, resulting in the C-peak.

Figure 2 shows the XRD patterns of Mg_77+*x*_Ni_20−*x*_La_3_ (*x* = 0, 5, 10, 15) alloys after saturated hydrogenation. It can be seen from the figure that the main phases of Mg_77_Ni_20_La_3_ alloy after saturated hydrogenation are Mg_2_NiH_4_, MgH_2_ and LaH_3.05_, which are manifested as peaks with different positions and intensities on the XRD pattern. As the Mg content increases and the Ni content decreases, the intensity of the Mg_2_NiH_4_ phase diffraction peak gradually decreases, while the intensities of the MgH_2_ and LaH_3.05_ phase diffraction peaks increase significantly. This indicates that the La_2_Mg_17_ and La_2_Mg_17_ phases before hydrogenation of the alloy are completely transformed into LaH_3.05_ and MgH_2_ phases, and no unmelted Mg and Ni elements are generated.

Figure 3 shows the scanning electron microscope (SEM) images of the cross-sections of the Mg_77+*x*_Ni_20−*x*_La_3_ (*x* = 0, 5, 10, 15) alloy ingots, showing the microstructure and phase composition of the alloys. As can be seen from the figure, the alloys are typical multiphase structures, represented by regions of different brightness. Combined with the EDS spectrum analysis of the corresponding regions in Figure 3a and the XRD test results, it can be known that the Mg_77_Ni_20_La_3_ alloy mainly consists of four phases: Mg_2_Ni, Mg, La_2_Mg_17_ and LaMg_12_, corresponding to the gray band-like regions, dark regions and bright continuous regions in the figure, respectively. The grey band in Figure 3a Mg_77_Ni_20_La_3_ is determined to be the Mg_2_Ni phase in combination with the phases identified by EDS spectroscopy and XRD in the corresponding region [51]. Similarly, the EDS energy spectrum analysis shows that the bright region is a coexistence of La_2_Mg_17_ and LaMg_12_ phases. In Figure 3b, Mg_82_Ni_15_La_3_, the long band-like Mg_2_Ni phase can be seen split into small pieces. The Mg phase gradually increases in the darker areas. The continuous bright areas begin to split into irregular polygons. Figure 3c The polygonal structure of the coexisting La_2_Mg_17_ and LaMg_12_ phases in the bright region of Mg_87_Ni_10_La_3_ is more regular. Figure 3d The microstructure and structure of the alloy changes considerably with increasing Mg content in Mg_92_Ni_5_La_3_. The grey and bright areas are uniformly distributed in the Mg matrix and show a typical eutectic structure consisting of Mg-Mg_2_Ni, a layered structure of several hundred nanometers [52]. This layered eutectic structure provides an efficient channel for the diffusion of hydrogen atoms during the hydrogen absorption and release process of the alloy [53]. These changes are consistent with the above XRD analysis results.

The XRD and SEM results show that the alloy decomposes into LaH_3.05_ and MgH_2_ after reacting with H_2_, which are formed from La_2_Mg_17_ and LaMg_12_ in the alloy. The XRD patterns of the hydrogenated alloy samples reveal that the diffraction peaks of the as-cast alloy phases disappear completely, and new peaks of metal hydrides emerge, including MgH_2_, LaH_3.05_ and Mg_2_NiH_4_. Thus, the first hydrogenation exothermic process of Mg_77+*x*_Ni_20−*x*_La_3_ (*x* = 0, 5, 10, 15) alloy can be expressed by the following chemical equations:Mg + H_2_ → MgH_2_(1)
Mg_2_Ni + H_2_ → Mg_2_NiH_4_(2)
LaMg_12_ + La_2_Mg_17_ + H_2_ → LaH_3.05_ + MgH_2_(3)

Moreover, after the dehydrogenation endothermic process, the MgH_2_ phase is completely transformed into Mg phase, but the rare earth hydride phase (LaH_3.05_) remains. This is because this kind of rare earth hydride has a high thermal stability [47], which cannot decompose under the current experimental conditions. Therefore, the reversible hydrogen absorption and desorption process of Mg_77+*x*_Ni_20−*x*_La_3_ (*x* = 0, 5, 10, 15) alloy after activation can be represented by the following two chemical equations:MgH_2_ ↔ Mg + H_2_(4)
Mg_2_NiH_4_ ↔ Mg_2_Ni + H_2_(5)

The stable LaH_3.05_ nanocrystals uniformly distributed on the surface and bulk of the alloy during hydrogen absorption and desorption create a large number of boundaries and active sites for hydrogen atom diffusion. This not only facilitates the nucleation of Mg/MgH_2_ and Mg_2_Ni/Mg_2_NiH_4_, but also enhances the diffusion rate of hydrogen atoms in the alloy. Meanwhile, the existence of Mg-MgH_2_ eutectic layered structure significantly increases the diffusion pathways of hydrogen atoms in the alloy. These two factors directly improve the kinetic properties of hydrogen absorption and desorption in the alloy.

### 3.2. Activation Behaviors and Hydrogen Absorption Kinetics

Figure 4 and Figure 5 shows the activation curves of Mg_77+*x*_Ni_20−*x*_La_3_ (*x* = 0, 5) alloy at 653 K. As shown in Figure 4a, the long-term hydrogen absorption curve of Mg_77_Ni_20_La_3_ alloy indicates that the alloy has an initial activation hydrogen absorption of 4.67 wt.%, which is much higher than the second and third activation hydrogen absorptions. In Figure 4b, the long-term hydrogen absorption curve of Mg_82_Ni_15_La_3_ alloy reveals that the alloy has an initial activation hydrogen absorption of 5.37 wt.%, while the second and third activation hydrogen absorptions are 5.16 wt.%, and the two curves almost overlap. This suggests that complete activation of the Mg_77_Ni_20_La_3_ alloy at 653 K requires 3 hydrogenation cycles, while the Mg_82_N_15_La_3_ alloy requires 2 hydrogenation cycles. In Figure 5a,b, the short-term hydrogen absorption curves of Mg_77_Ni_20_La_3_ and Mg_82_Ni_15_La_3_ alloys demonstrate that Mg_77_Ni_20_La_3_ alloy takes much less time than Mg_82_Ni_15_La_3_ alloy to reach the saturated hydrogen absorption (90%) in the second and third activations. This implies that Mg_77_Ni_20_La_3_ alloy has a better activation performance than Mg_82_Ni_15_La_3_ alloy. Moreover, it can also be seen from the short-term activation curves that the initial activation of the alloy is very slow.

Figure 6 shows the hydrogen absorption kinetics curves of Mg_77+*x*_Ni_20−*x*_La_3_ (*x* = 0, 5) alloy at different temperatures. As shown in Figure 6a, the hydrogen absorption of Mg_77_Ni_20_La_3_ alloy reaches 4.02 wt.% at 533 K for 2 h, and the hydrogen absorption kinetics curves of Mg_77_Ni_20_La_3_ alloy are almost identical when the temperature is above 533 K, indicating excellent hydrogen absorption kinetics properties. The hydrogen absorption at 623 K and 683 K are 4.3 wt.% and 4.16 wt.%, respectively. Figure 6b reveals that the hydrogen absorption of Mg_82_Ni_15_La_3_ alloy attains as high as 4.92 wt.% at 683 K for 2 h. This is because the higher the Mg content, the stronger the hydrogen absorption ability of the alloy. The hydrogen absorption amount of the initial activation curve in Mg_77_Ni_20_La_3_ and Mg_82_Ni_15_La_3_ alloys is significantly higher than that of the alloy at each temperature, which is due to the formation of LaH_3.05_ phase with high thermal stability and difficult to decompose after the initial hydrogenation of the alloy. In summary, Mg_82_Ni_15_La_3_ alloy has better hydrogen absorption kinetics performance than Mg_77_Ni_20_La_3_ alloy.

### 3.3. Hydrogen Desorption Kinetics and Activation Energy

Figure 7 and Figure 8 show the hydrogen desorption endothermic reaction kinetics curves of Mg_77+*x*_Ni_20−*x*_La_3_ (*x* = 0, 5) alloys at different temperatures. It can be seen from Figure 7a that Mg_77_Ni_20_La_3_ alloy has the best hydrogen desorption working temperature above 623 K and reaches a hydrogen desorption amount of 4.1 wt.% at 683 K. It can be seen from Figure 7b that Mg_82_Ni_15_La_3_ alloy has a hydrogen desorption amount of up to 4.69 wt.% at 683 K. This is because the higher the Mg content, the stronger the hydrogen desorption ability of the alloy. It can be seen from Figure 8a that the short-term hydrogen desorption endothermic kinetics curve of Mg_77_Ni_20_La_3_ alloy shows that the maximum hydrogen desorption amount of the alloy is 3.8 wt.% at 653 K and reaches the saturated hydrogen desorption amount within 125 s. As the temperature increases, it only takes 71 s to reach a maximum hydrogen desorption amount of 3.8 wt.% at 683 K. This is mainly due to the increase of temperature, which increases the energy of reactants in the alloy, accelerates the hydrogen desorption rate of the alloy, and thus increases the hydrogen desorption amount. At the same time, Figure 8b shows that the short-term hydrogen desorption exothermic kinetics curve of Mg_82_Ni_15_La_3_ alloy shows that the alloy reaches a saturated hydrogen desorption amount of 4.39 wt.% within 140 s at 653 K. And when the temperature rises to 683 K, Mg_82_Ni_15_La_3_ alloy reaches a hydrogen desorption amount of 3.8 wt.% within 47 s.

The phase change heat of the alloy was calculated by using the test method of the DSC curve of the alloy hydride warming and exothermic reaction. In order to calculate the phase transition heat of the alloy, we adopted the method of testing the DSC curve of the hydrogen desorption endothermic of the alloy hydride. Figure 9 shows the DSC curves of Mg_77+*x*_Ni_20−*x*_La_3_ (*x* = 0, 5, 10, 15) alloy hydrides after saturated hydrogen absorption at different heating rates. The test temperature range was 400~700 K. Firstly, the initial temperature of the hydrogen desorption endothermic reaction of the alloy can be obtained from the DSC curve, and the activation energy of the hydrogen desorption endothermic reaction can be calculated according to the Kissinger method, so as to judge the difficulty of the reaction. It can be seen from Figure 9 that there is a clear exothermic main peak and a smaller peak on each DSC curve at each heating rate. The smaller exothermic peak is related to the phase transition of Mg_2_NiH_4_, by the coupled hydrogen desorption endothermic of MgH_2_ and Mg_2_NiH_4_ phases. The temperature range of the smaller exothermic peak is between 513–522 K. In Figure 9a, the endothermic main peak consists of two obvious peaks, which confirms that the exothermic main peak is caused by two hydrogen desorption phases of MgH_2_ and Mg_2_NiH_4_, while in Figure 9b–d, the exothermic main peak becomes an obvious peak, because two kinds of hydrogen desorption phases have a synergistic hydrogen desorption endothermic phenomenon. In Figure 9e, the exothermic peak temperature at 20 K/min heating rate shows that the phase transition peak temperature of Mg_2_NiH_4_ is little affected by Mg content and remains around 520 K. While the peak temperature of MgH_2_ is more affected by Mg content and shows a trend of first rising and then falling. The increase of hydrogen desorption peak temperature causes the increase of activation energy of hydrogen desorption endothermic reaction of alloy hydride, which will be calculated in detail below. By comparing the hydrogen desorption endothermic temperatures corresponding to the exothermic main peaks of four alloys with different Mg contents at the same rate, it is found that with the increase of Mg content, the hydrogen desorption peak temperature of the alloy gradually increases and then decreases. Mg_87_Ni_10_La_3_ alloy has the highest hydrogen desorption peak temperature at 20 K/min heating rate, which is 629.8 K; while Mg_92_Ni_5_La_3_ alloy is 624.1 K, which is 5.7 K lower.

Table 1 lists the peak temperatures of DSC endothermic dehydrogenation reaction of Mg_77+*x*_Ni_20−*x*_La_3_ (*x* = 0, 5, 10, 15) alloy hydrides at different heating rates. Using different heating rates and corresponding peak temperatures, the activation energy (Ea) of dehydrogenation endothermic reaction of the alloy can be calculated by Kissinger method.

According to the Kissinger method, the activation energy can be calculated by using the following Equation (6):(6)$$\ln\left(\frac{\beta }{T_{p}^{2}}\right)=\frac{-E_{a}}{RT_{P}}+\;\ln\left(k_{0}\right)$$
where *R* is the gas constant, *k*_0_ is a constant, *T_p_* is the peak temperature of the exothermic peak, and *β* is the heating rate. This method was used to calculate the activation energy of Mg_77+*x*_Ni_20−*x*_La_3_ (*x* = 0, 5, 10, 15) alloys at different heating rates.

Figure 10 shows the curve diagram after fitting according to the Kissinger equation. The fitting results show that the dehydrogenation activation energy of Mg_82_Ni_15_La_3_ alloy is (72.7 ± 4.1) kJ/mol, the dehydrogenation activation energy of Mg_87_Ni_10_La_3_ alloy is larger, which is (75.2 ± 1.5) kJ/mol, and the dehydrogenation activation energy of Mg_92_Ni_5_La_3_ alloy is smaller, which is (68.5 ± 3.8) kJ/mol. Compared with Mg_87_Ni_10_La_3_ alloy, the dehydrogenation endothermic activation energy of Mg_92_Ni_5_La_3_ alloy is reduced by about 9%. Figure 10 shows that the activation energy values of Mg_77_Ni_20_La_3_, Mg_82_Ni_15_La_3_, Mg_87_Ni_10_La_3_ and Mg_92_Ni_5_La_3_ are between 68.5~75.2 kJ/mol. After comparison, these values are obviously lower than (150~160 kJ/mol) of MgH_2_. The reason for the low activation energy value of the alloy is the effect of alloying and rare earth hydride formed by adding rare earth elements. In addition, due to the addition of rare earth element La to form rare earth hydride LaH_3.05_ phase, and LaH_3.05_ phase has “hydrogen pumping”, which greatly reduces the hydrogen activation energy value of the alloy.

The increase of dehydrogenation temperature and dehydrogenation activation energy of alloy hydride is not conducive to the improvement of dehydrogenation kinetics performance of alloy. In Mg_77+*x*_Ni_20−*x*_La_3_ (*x* = 0, 5, 10, 15) alloy, Mg content increases from Mg_77_ to Mg_92_, and Ni content decreases from Ni_20_ to Ni_5_. It can be seen from the hydrogen absorption and dehydrogenation kinetics curves of the alloy that the hydrogen absorption amount of the alloy shows an increasing trend. This is because after the increase of Mg content, the hydrogen absorption and desorption ability of Mg phase is higher than that of Mg_2_Ni phase. And with the increase of hydrogen absorption and desorption cycle times, the hydrogen absorption and desorption ability of the alloy does not weaken obviously, and the alloy can achieve complete hydrogen absorption and desorption.

The peak temperature of hydrogen desorption endothermic reaction and phase transition heat for Mg_77+*x*_Ni_20−*x*_La_3_ (*x* = 0, 5, 10, 15) alloy is presented in Table 2. The phase change heats are based on the DSC curves of the exothermic hydrides of the alloy after saturated hydrogen absorption at 5, 10, 20 and 40 K/min heating rates, calculated by the phase change heat (area of the peak) calculation function that comes with the DSC software. From the table, it can be seen that the phase change heat of Mg_77_Ni_20_La_3_ alloy is 1177 J/g at a heating and desorption hydrogen rate of 5 K/min; the phase change heat of Mg_87_Ni_10_La_3_ alloy is 1529 J/g; the phase change heat of Mg_92_Ni_5_La_3_ alloy is 1428 J/g, which is 251 J/g higher than that of Mg_77_Ni_20_La_3_ alloy. The phase change heat of Mg_92_Ni_5_La_3_ alloy is second only to Mg_87_Ni_10_La_3_ alloy, while the peak hydrogen desorption temperature is significantly lower than that of Mg_87_Ni_10_La_3_, which is more favorable to the hydrogen desorption and heat absorption reaction of the alloy, but due to the high Mg content, it will lead to the alloy not releasing hydrogen. It can be seen from the analysis that with the increase of Mg content, the peak endothermic temperature of the alloy gradually increases, and the corresponding phase transition heat also gradually increases. It can be seen that the phase transition heat of Mg_87_Ni_10_La_3_ alloy is significantly higher than that of the first two alloys and slightly higher than that of Mg_92_Ni_5_La_3_ alloy. At the same time, the higher the Mg content in the alloy, the more likely the alloy will not desorption hydrogen. In summary, Mg_87_Ni_10_La_3_ alloy is more suitable as an energy storage material.

## 4. Conclusions

The Mg_77+*x*_Ni_20−*x*_La_3_ (*x* = 0, 5, 10, 15) alloy prepared by induction melting method mainly consists of three main phases: Mg_2_Ni, LaMg_12_, La_2_Mg_12_ and residual Mg phase. After saturated hydrogen absorption, Mg_2_Ni phase transforms into MgH_2_ and LaH_3.05_ phase. With the increase of Mg content, the long strip-like Mg_2_Ni phase gradually disappears and transforms into Mg-Mg_2_Ni eutectic lamellar structure. Mg_77+*x*_Ni_20−*x*_La_3_ (*x* = 0, 5, 10, 15) alloys can be fully activated after 3 hydrogenation cycles at 653 K. The hydrogen absorption capacity of Mg_77_Ni_20_La_3_ and Mg_82_Ni_15_La_3_ alloys at 683 K for 2h are 4.92 wt.% and 4.69 wt.%, respectively. The hydrogen desorption activation energy of Mg_77_Ni_20_La_3_ alloy is (70.9 ± 4.2) kJ/mol, Mg_82_Ni_15_La_3_ alloy is (72.7 ± 4.1) kJ/mol, Mg_87_Ni_10_La_3_ alloy is (75.2 ± 1.5) kJ/mol, and Mg_92_Ni_5_La_3_ alloy is (68.5 ± 3.8) kJ/mol. The DSC test results show that the phase transition heat of Mg_82_Ni_15_La_3_ alloy reaches 1529 J/g, which is 352 J/g higher than that of Mg_77_Ni_20_La_3_ alloy.

## Figures and Tables

**Figure 1 materials-16-04576-f001:**
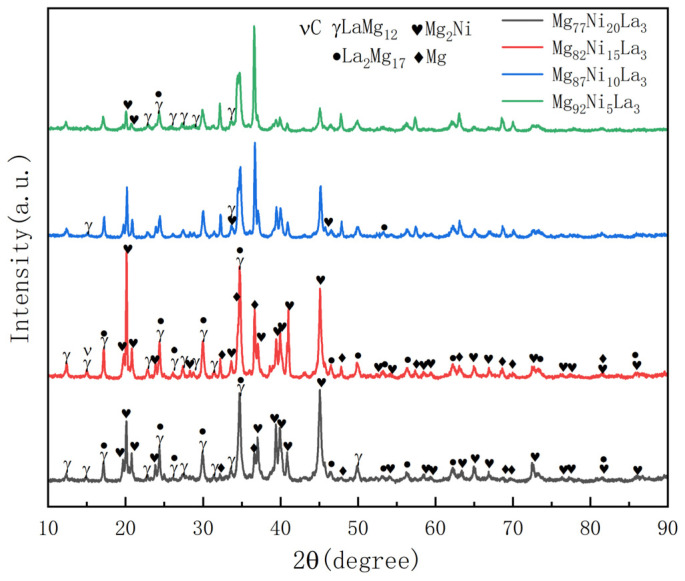
XRD patterns of the as-cast Mg_77+*x*_Ni_20−*x*_La_3_ (*x* = 0, 5, 10, 15) alloys.

**Figure 2 materials-16-04576-f002:**
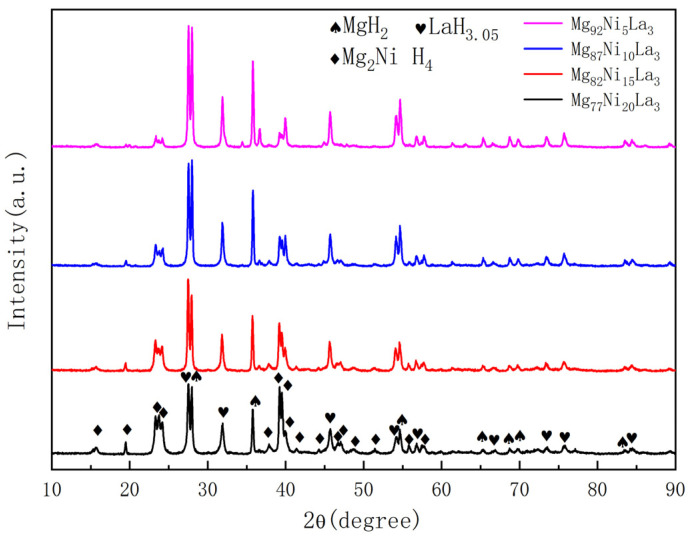
XRD patterns of the full hydrogenated Mg_77+*x*_Ni_20−x_La_3_ (*x* = 0, 5, 10, 15) alloys.

**Figure 3 materials-16-04576-f003:**
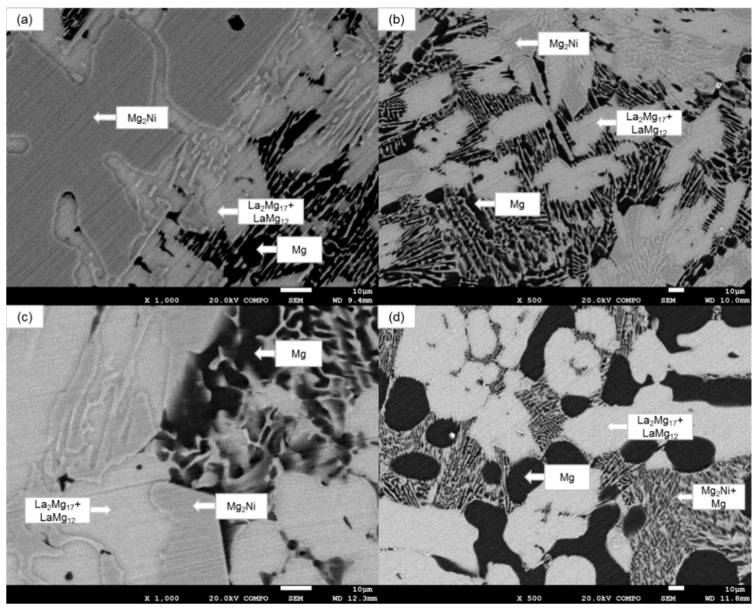
SEM images of the as-cast alloy ingots: (**a**) Mg_77_Ni_20_La_3_; (**b**) Mg_82_Ni_15_La_3_; (**c**) Mg_87_Ni_10_La_3_, (**d**) Mg_92_Ni_5_La_3_.

**Figure 4 materials-16-04576-f004:**
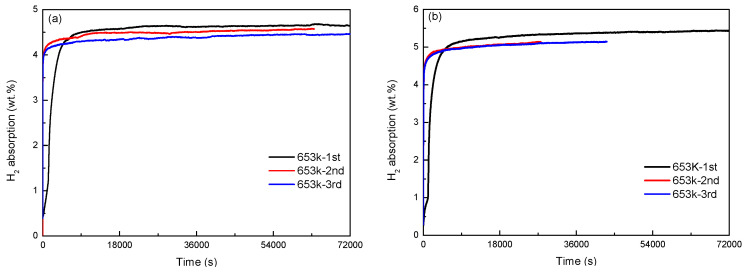
Long-time activation curves of Mg_77+*x*_Ni_20−*x*_La_3_ (*x* = 0, 5) alloys at 653 K: (**a**) long-time hydrogen absorption by Mg_77_Ni_20_La_3_; (**b**) long-time hydrogen absorption by Mg_82_Ni_15_La_3_.

**Figure 5 materials-16-04576-f005:**
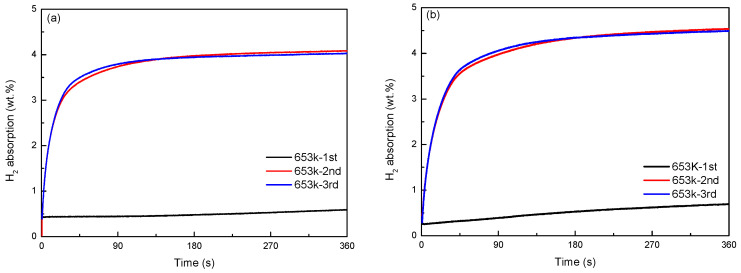
Short-time activation curves of Mg_77+*x*_Ni_20−*x*_La_3_ (*x* = 0, 5) alloys at 653 K: (**a**) short-time hydrogen absorption by Mg_77_Ni_20_La_3_; (**b**) short-time hydrogen absorption by Mg_82_Ni_15_La_3_.

**Figure 6 materials-16-04576-f006:**
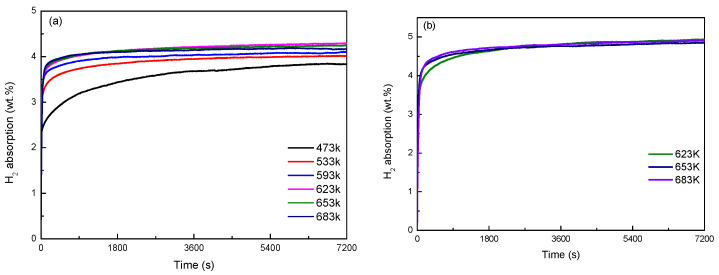
Hydrogen absorption kinetic curves of Mg_77+*x*_Ni_20−*x*_La_3_ (*x* = 0, 5) alloys at various temperatures: (**a**) Mg_77_Ni_20_La_3_; (**b**) Mg_82_Ni_15_La_3_.

**Figure 7 materials-16-04576-f007:**
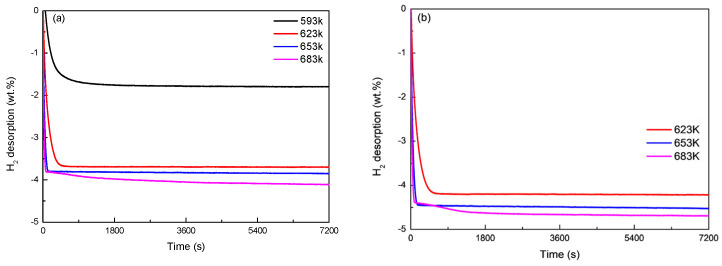
Long-time hydrogen desorption kinetic curves of Mg_77+*x*_Ni_20−*x*_La_3_ (*x* = 0, 5) alloys at 653 K: (**a**) long-time hydrogen desorption by Mg_77_Ni_20_La_3_; (**b**) long-time hydrogen desorption by Mg_82_Ni_15_La_3_.

**Figure 8 materials-16-04576-f008:**
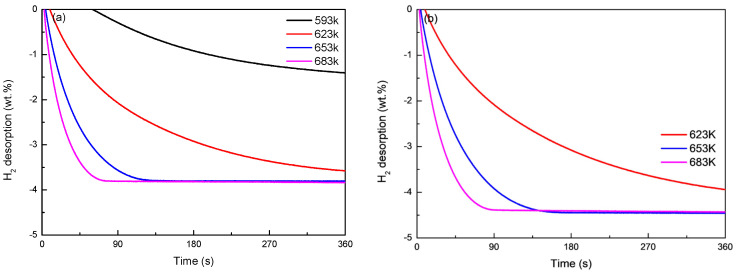
Short-time hydrogen desorption kinetic curves of Mg_77+*x*_Ni_20−*x*_La_3_ (*x* = 0, 5) alloys at 653 K: (**a**) short-time hydrogen desorption by Mg_77_Ni_20_La_3_; (**b**) short-time hydrogen desorption by Mg_82_Ni_15_La_3_.

**Figure 9 materials-16-04576-f009:**
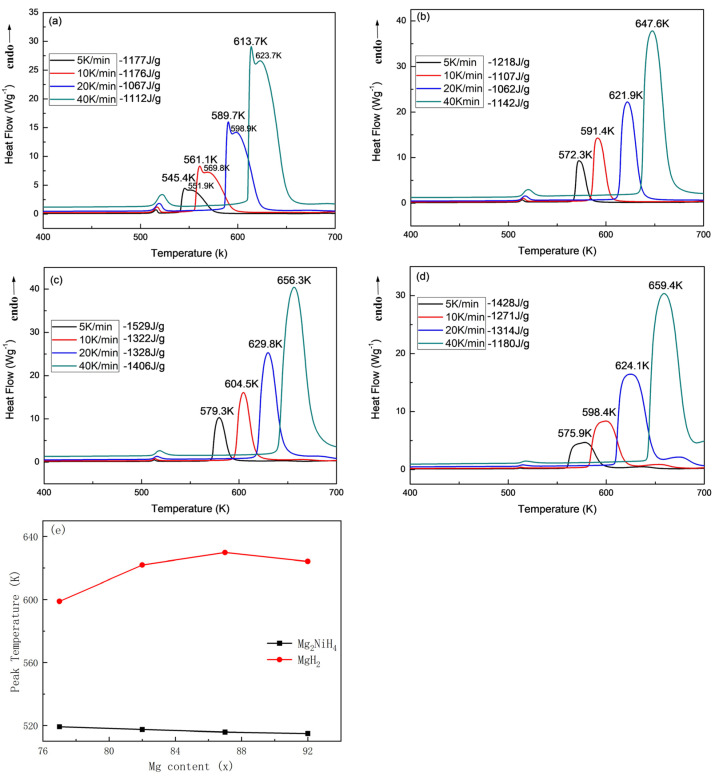
DSC curves of saturated hydrides of Mg_77+*x*_Ni_20−*x*_La_3_ (*x* = 0, 5, 10, 15) alloys: (**a**) Mg_77_Ni_20_La_3_; (**b**) Mg_82_Ni_15_La_3_; (**c**) Mg_87_Ni_10_La_3_; (**d**) Mg_92_Ni_5_La_3_; (**e**) two peak heat absorption temperatures with increase of Mg content at 20 K/min heating rate contention.

**Figure 10 materials-16-04576-f010:**
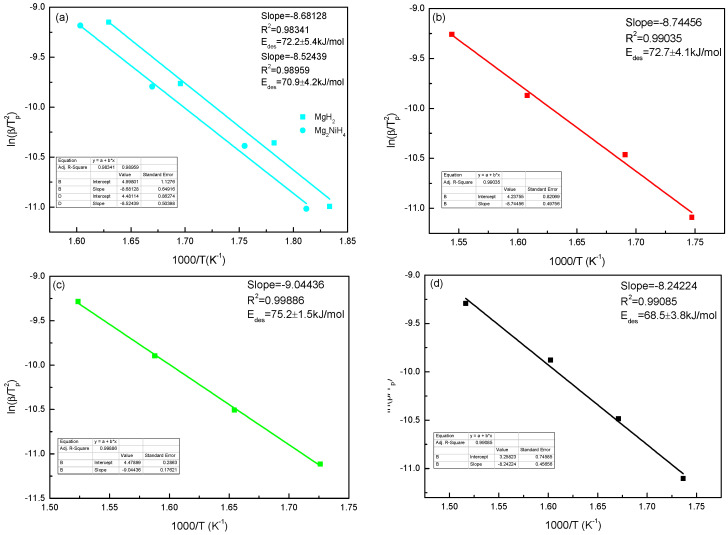
Kissinger plot corresponding peak temperature of DSC at different heating rates of Mg_77+*x*_Ni_20−*x*_La_3_ (*x* = 0, 5, 10, 15) alloys: (**a**) Mg_77_Ni_20_La_3_; (**b**) Mg_82_Ni_15_La_3_; (**c**) Mg_87_Ni_10_La_3_; (**d**) Mg_92_Ni_5_La_3_.

**Table 1 materials-16-04576-t001:** The peak temperature of Mg_77+*x*_Ni_20−*x*_La_3_ (*x* = 0, 5, 10, 15) alloy hydrides during DSC dehydrogenation reaction at different heating rates.

Alloy	5 K/min		10 K/min		20 K/min		40 K/min	
	Mg_2_NiH_4_	MgH_2_	Mg_2_NiH_4_	MgH_2_	Mg_2_NiH_4_	MgH_2_	Mg_2_NiH_4_	MgH_2_
Mg_77_Ni_20_La_3_	515.7	551.9	516.6	569.8	519.2	598.9	522.3	623.7
Mg_82_Ni_15_La_3_	514.4	572.3	514.9	591.4	517.5	621.9	520.5	647.6
Mg_87_Ni_10_La_3_	513.2	579.3	514.8	604.5	515.8	629.8	518.4	656.3
Mg_92_Ni_5_La_3_	513.6	575.9	513.6	598.4	514.9	624.1	517.5	659.4

**Table 2 materials-16-04576-t002:** The peak temperature of hydrogen desorption endothermic reaction and phase transition heat for Mg_77+*x*_Ni_20−*x*_La_3_ (*x* = 0, 5, 10, 15) alloy.

Alloy	Peak Temperature/°C				Melting Enthalpy/J·g^−1^			
	5 K/min	10 K/min	20 K/min	40 K/min	5 K/min	10 K/min	20 K/min	40 K/min
Mg_77_Ni_20_La_3_	545.4	561.1	589.7	613.7	−1177	−1176	−1067	−1112
Mg_82_Ni_15_La_3_	572.3	591.4	621.9	647.6	−1218	−1107	−1062	−1142
Mg_87_Ni_10_La_3_	579.3	604.5	629.8	656.3	−1529	−1322	−1328	−1406
Mg_92_Ni_5_La_3_	575.9	598.4	624.1	659.4	−1428	−1271	−1314	−1180

## Data Availability

Due to privacy and ethical restrictions, the data supporting the reported results of this study are unavailable for sharing. We acknowledge the importance of data sharing and encourage researchers to comply with data availability policies.

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
