# Peer review of "Microstructure Characteristics and Hydrogen Storage Kinetics of Mg77+xNi20−xLa3 (x = 0, 5, 10, 15) Alloys"

_materials, 2023, doi:10.3390/ma16134576_

Round 1

Reviewer 1 Report

  1. Authors should elaborate on the specific techniques and methodologies used for the structural characterizations of the Mg77+xNi20-xLa3 alloys, such as X-ray diffraction (XRD) and scanning electron microscopy (SEM)? How were these techniques applied to analyze the microstructure and phase composition of the alloys?
  2. Provide a more comprehensive explanation of the observed effects of Nickel (Ni) content on the microstructure of the as-cast Mg77+xNi20-xLa3 alloys? How does the variation in Ni content influence the distribution and characteristics of different phases within the alloys?
  3. Please provide more detailed descriptions of the phases present in the Mg77+xNi20-xLa3 alloys before hydrogenation. Specifically, what are the characteristics and compositions of the LaMg12, Mg2Ni, La2Mg17, and Mg phases observed in the alloys?
  4. How does the increase in Mg content affect the intensities of the LaMg12, Mg2Ni, and La2Mg17 phases in the Mg77+xNi20-xLa3 alloys? Could you explain the relationship between Mg content and the relative amounts of these phases in the alloys?
  5. Authors should elaborate further on the role of grain boundaries in facilitating hydrogen diffusion within the Mg77+xNi20-xLa3 alloys? How do the grain boundaries contribute to the improved kinetic properties of hydrogen absorption and desorption in the alloys?
  6. What are the specific factors responsible for the presence of the C peak observed in the XRD pattern of the Mg77+xNi20-xLa3 alloys? How does the graphite crucible used during alloy preparation contribute to the formation of this peak?
  7. Provide more in-depth explanations of the phases present in the Mg77+xNi20-xLa3 alloys after saturated hydrogenation? How does the hydrogenation process transform the Mg2Ni, Mg, La2Mg17, and LaMg12 phases into MgH2, LaH3.05, and other hydrogenated phases?
  8. How do variations in Mg and Ni content influence the intensities of the Mg2NiH4, MgH2, and LaH3.05 phases observed after hydrogenation in the Mg77+xNi20-xLa3 alloys? Can you explain the relationship between the alloy composition and the formation of these hydrogenated phases?
  9. Provide more detailed descriptions and explanations of the observed microstructure characteristics in the cross-sections of the Mg77+xNi20-xLa3 alloy ingots? How do the different regions of brightness correspond to the different phases present in the alloys?
  10. Based on the SEM images and EDS spectrum analysis, can you further elaborate on the specific phases constituting the Mg77Ni20La3 alloy? How do the gray band-like regions, dark regions, and bright continuous regions in the SEM images correspond to the Mg2Ni, Mg, La2Mg17, and LaMg12 phases, respectively?
  11. How do the microstructure and morphology of the Mg82Ni15La3 alloy differ from those of the Mg77Ni20La3 alloy? Can you provide more specific details about the changes observed in the dark regions, bright continuous regions, and overall microstructure?
  12. Provide more information and explanations regarding the significant microstructure and morphology changes observed in the Mg92Ni5La3 alloy compared to the other alloys? How do the dispersed ellipses and separation of La2Mg17 and LaMg12 phases manifest in the alloy's microstructure?
  13. Based on the study findings, what are the main conclusions regarding the phase composition and transformation behaviors of the Mg77+xNi20-xLa3 alloys? Can you summarize the primary phases present in the alloys before and after hydrogenation?
  14. Can you explain in more detail how the Mg2Ni phase transforms into MgH2 and LaH3.05 phases during saturated hydrogen absorption in the Mg77+xNi20-xLa3 alloys? What are the mechanisms and kinetics involved in this phase transformation process?
  15. How is the Mg-Mg2Ni eutectic lamellar structure formed as the Mg content increases in the Mg77+xNi20-xLa3 alloys? Could you provide more insights into the formation mechanism and the role of this lamellar structure in enhancing hydrogen diffusion within the alloys?
  16. Authors should clarify the number of hydrogenation cycles required to fully activate the Mg77+xNi20-xLa3 alloys at 653K? How does the activation process impact the hydrogen absorption and desorption properties of the alloys?
  17. What are the precise hydrogen absorption capacities of the Mg77Ni20La3 and Mg82Ni15La3 alloys at 683K for a duration of 2 hours? Can you provide the corresponding weight percentages of absorbed hydrogen for each alloy?
  18. Provide more precise values for the hydrogen desorption activation energy among the different Mg77+xNi20-xLa3 alloys? How do these values compare to the range of 68.5~75.2 kJ/mol calculated using the Kissinger equation?
  19. explain the significance of the observed phase transition heat of the Mg82Ni15La3 alloy, which is 352J/g higher than that of the Mg77Ni20La3 alloy? What implications does this have for the hydrogen storage properties of the alloys?
  20. Based on the improved hydrogen-absorption kinetic performance of the Mg77+xNi20-xLa3 alloys, what potential applications can be envisioned for these alloys in the field of hydrogen storage? How do the findings of this study contribute to the development of more efficient hydrogen storage systems?
  21. What are the potential applications of the Mg77+xNi20-xLa3 alloys based on their hydrogen storage properties and improved kinetic performance? How do these alloys compare to existing hydrogen storage materials in terms of their performance and practicality?
  22. Can you provide more details about the X-ray diffraction (XRD) analysis performed to analyze the phase composition of the Mg77+xNi20-xLa3 alloys? What specific parameters and techniques were used during the XRD measurements?
  23. What is the specific role and significance of the LaMg12 and La2Mg12 phases in the Mg77+xNi20-xLa3 alloys? How do these phases contribute to the overall hydrogen storage properties and behavior of the alloys?
  24. How does the hydrogen absorption capacity of the Mg77+xNi20-xLa3 alloys vary with increasing Mg content? Can you provide more insights into the relationship between Mg content and the hydrogen absorption properties of the alloys?
  25. Can you explain in more detail how the presence of the Mg-Mg2Ni eutectic lamellar structure enhances the diffusion of hydrogen atoms within the Mg77+xNi20-xLa3 alloys? What specific mechanisms or structural characteristics facilitate this enhanced diffusion? Considering the results obtained, what are the prospects for the Mg77+xNi20-xLa3 alloys to be considered as potential hydrogen storage materials? How do their properties and performance compare to other existing materials?

Minor editing of English language required. 

Author Response

Thank you for your comments concerning our manuscript entitled “Microstructure characteristics and hydrogen storage kinetics of Mg77+xNi20-xLa3 (x=0,5,10,15) alloys” (ID: materials-2437109). Those comments are all valuable and very helpful for revising and improving our paper, as well as the important guiding significance to our researches. We have studied comments carefully and have made corrections which we hope meet with approval. All the revised portions have been highlighted by blue colour.

Reviewer 2 Report

This manuscript investigates the effects of nickel content on the microstructure and hydrogen storage kinetic of the as-cast alloys Mg77+xNi20-xLa3 (x=0,5,10,15). This MS provides new and interesting insights into the important problem of finding a material that can be a potential for solid-state hydrogen storage and battery. I would suggest the publication in the Materials with revisions as follows.

1)      In Abstract, please delete abbreviation for Ea. NEVER state the abbreviation unless it is used later in the abstract.

2)      All the figures are too small. Please enlarged.

3)      Please mark the endothermic/exothermic sign (arrow) in the Fig. 9(a), 9(b), 9(c) and 9(d).

4)      It is informative if the authors can show the XRD curve after dehydrogenated process.

5)      The authors report the results of DSC measurements. It would also be interesting to report the enthalpy change of Mg77+xNi20-xLa3 (x=0,5,10,15). It is a piece of information easy to extract from the DSC curves already carried out.

6)      There are several papers that related with Mg-based materials for solid-state hydrogen storage recently published in Materials should be refer or cite in introduction part or result and discussion part such as Materials, 16(6) (2023) 2526; Materials, 15(22) (2022) 8126; Materials, 15(22) (2022) 1667; Materials, 16(6) (2023) 2449; Materials, 16(6) (2023) 2176

7)      Write the conclusion more precious.

8)      Please check the typing and grammatical error for the whole manuscript.

Minor editing of English language required

Author Response

(The authors gave the same response as above.)

Reviewer 3 Report

The subject of this paper has been visited many times, since the discovery of Mg2NiH4 related hydrides for hydrogen storage. Yet another paper makes not much diffference.

Mg2NiH4 has also interesting electric conductivity issues related to the low to high temperature phase transition, but that is not treated in the article.

Author Response

Thank you for yourcomments concerning our manuscript entitled “Microstructure characteristics and hydrogen storage kinetics of Mg77+xNi20-xLa3 (x=0,5,10,15) alloys” (ID: materials-2437109). Those comments are all valuable and very helpful for revising and improving our paper, as well as the important guiding significance to our researches. We have studied comments carefully and have made corrections which we hope meet with approval. All the revised portions have been highlighted by blue colour. The main corrections in the paper and responds to the reviewer’s comments are as follows:

Composion of the alloys for our paper has not been investigated by other researchers.

Round 2

Reviewer 1 Report

The revised manuscript can be accepted. 

Minor editing of English language is required.